# Deeply Shared Filter Bases for Parameter-Efficient Convolutional Neural Networks

**Woochul Kang**
Department of Embedded Systems Engineering
Incheon National University
Yeonsu-gu, South Korea 22012
wchkang@inu.ac.kr

**Daeyeon Kim**[*]
Dain Technology, Inc.
dustin@soundablehealth.com

## Abstract

Modern convolutional neural networks (CNNs) have massive identical convolution blocks, and, hence, recursive sharing of parameters across these blocks has been proposed to reduce the amount of parameters. However, naive sharing of parameters poses many challenges such as limited representational power and the vanishing/exploding gradients problem of recursively shared parameters. In this paper, we present a recursive convolution block design and training method, in which a recursively shareable part, or a filter basis, is separated and learned while effectively avoiding the vanishing/exploding gradients problem during training. We show that the unwieldy vanishing/exploding gradients problem can be controlled by enforcing the elements of the filter basis orthonormal, and empirically demonstrate that the proposed *orthogonality regularization* improves the flow of gradients during training. Experimental results on image classification and object detection show that our approach, unlike previous parameter-sharing approaches, does not trade performance to save parameters and consistently outperforms overparameterized counterpart networks. This superior performance demonstrates that the proposed recursive convolution block design and the orthogonality regularization not only prevent performance degradation, but also consistently improve the representation capability while a significant amount of parameters are recursively shared.

## 1 Introduction

Modern convolutional neural networks (CNNs) such as ResNets have massive identical convolution blocks and recent analytic studies (Jastrzebski et al., 2018) show that these blocks perform mostly iterative refinement of features rather than learning new features. Inspired by these results, recursive sharing of weights has been studied as a promising direction to parameter-efficient CNNs (Jastrzebski et al., 2018; Guo et al., 2019; Savarese & Maire, 2019). However, naive sharing of parameters across many convolution layers incurs several problems. First of all, recursive sharing of parameters can result in the *vanishing* and the *exploding gradients* problem, which is one of the main reasons that recurrent neural networks (RNNs) are so hard to train properly (Pascanu et al., 2013; Jastrzebski et al., 2018). Another problem is that overall representation power can be limited by iterative sharing of parameters. Due to these challenges, most compression approaches based on parameter-sharing suffer from performance degradation.

In this work, we conjecture that convolution layers or blocks can be separated into inherently shareable parts and non-shareable parts, and can be trained effectively by avoiding the vanishing and the exploding gradients problem. To achieve this, for a full convolution operator, we first replace it with a factorized version that splits the convolution operator into two separate operators; one

---

[*]W. Kang and D. Kim are contributed equally. This work was conducted at Incheon National University.

operator with inherently shareable filters, called a *filter basis*, and the other operator with non-shared filters, called *coefficients*. When successive convolution blocks share a common filter basis, they are positioned in the same vector subspace. However, their representation capability is retained through non-shared coefficients that learn diverse features by linearly combining the shared filter basis.

By separating shareable parts from non-shareable parts, we can impose desirable properties on the shared parameters. To avoid performance degradation from recursive sharing of parameters, we propose the *orthogonality regularization*, in which the vanishing/exploding gradients problem is controlled by enforcing the elements of a shared filter basis orthonormal during training. We both theoretically and empirically show that the proposed orthogonality regularization improves the flow of the gradients during training and reduces the redundancy in parameters effectively.

For efficient CNNs such as MobileNets (Howard et al., 2017), we do not need to factorize convolution operators to uncover a shared filter basis since these networks already have factorized convolution block structures for computational efficiency. For such networks, our approach can be applied simply by identifying one or two convolution operators of repeating convolution blocks as a filter basis that shares weights across the repeating blocks. Other convolution operators in each convolution block become block-specific non-shared coefficients.

Since our focus is not on pushing the state-of-the-art performance, we demonstrate the effectiveness of our work using widely-used models as base models on image classification and object detection. Without bells and whistles, simply applying the proposed convolution block design and the orthogonality regularization saves a significant amount of parameters while consistently outperforming over-parameterized counterpart networks. For example, our method can save up to 46.0% of parameters of ResNets while consistently achieving lower test errors. Even in compact models, such as MobileNetV2, our approach can achieve further 8-21% parameter savings while outperforming the original models. This superior performance demonstrates that the proposed recursive convolution blocks and orthogonality regularization enables effective learning of better feature representations while a significant amount of parameters are shared recursively.

## 2 Related Work

**Recursive networks and parameter sharing:** Recurrent neural networks (RNNs) (Graves et al., 2013) have been well-studied for temporal and sequential data. As a generalization of RNNs, recursive variants of CNNs are used extensively for visual tasks (Socher et al., 2011; Liang & Hu, 2015; Xingjian et al., 2015; Kim et al., 2016; Zamir et al., 2017). For instance, Eigen et al. (2014) explore recursive convolutional architectures that share filters across multiple convolution layers. They show that recurrence with deeper layers tends to increase performance. However, their recursive architecture shows worse performance than independent convolution layers due to overfitting. In contrast, we share only fundamentally shareable parts, or filter bases, that are separated from conventional convolution operators. By separating shareable parts from non-shareable parts, we can impose desirable properties on the shared parameters that prevents the vanishing/exploding gradients problems without damaging representational capability of the networks.

More recently, Jastrzebski et al. (2018) show that iterative refinement of features in ResNets suggests that deep networks can potentially leverage intensive parameter sharing. Guo et al. (2019) introduce a gate unit to determine whether to jump out of the recursive loop of convolution blocks to save computational resources. These works show that training recursive networks with naively shared blocks leads to bad performance due to the problem of gradient explosion and vanish like RNN (Pascanu et al., 2013; Vorontsov et al., 2017). In order to deal with the problem of gradient explosion and vanish, they suggest unshared batch normalization strategy. In our work, we propose the orthogonality regularization of shared filter bases to further address this problem.

A few recent works generate convolution filters by combining shared building blocks (Bhalgat et al., 2020; Savarese & Maire, 2019; Yang et al., 2019; Qiu et al., 2018). Although these approaches are different in details, they are similar to our work since they all share a set of filters and combine them to build layer-specific filters. However, they save parameters at the cost of accuracy loss. For example, in ResNet50 on ImageNet, Yang et al. (2019)'s work incurs 0.9% accuracy drop, while our method achieves performance improvement. We believe that this performance gap comes from the gradients issue of shared parameters and these previous works could benefit more from the improved flow of gradients of our approach.

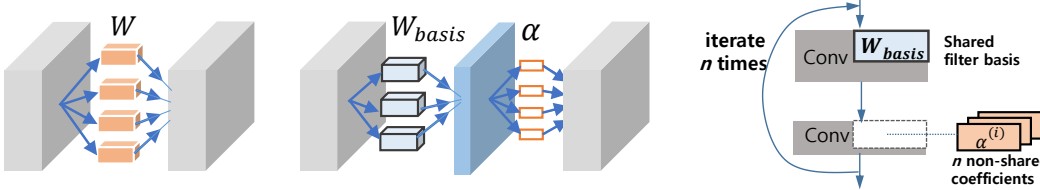

(a) Full convolution         (b) A filter basis and coefficients         (c) Iterative sharing of a filter basis

Figure 1: A full convolution operator in (a) with a filter $W$ can be replaced by two separate convolution operators, as in (b): one operator with a *filter basis* ($W_{basis}$) and the other operator with *coefficients* ($\alpha$s). A filter basis is inherently shareable for convolution operators in the same vector subspace, and, hence, a recursive architecture is suggested, as in (c). The filter basis shared by recursive convolution layers can be learned through typical gradient-based training while avoiding potential vanishing/exploding gradients problem by enforcing its elements orthonormal.

**Model compression and efficient convolution block design:** Reducing storage and inference time of CNNs has been an important research topic for both resource constrained mobile/embedded systems and energy-hungry data centers. A number of research techniques have been developed such as filter pruning (LeCun et al., 1990; Polyak & Wolf, 2015; Li et al., 2017; He et al., 2017b), low-rank factorization (Denton et al., 2014; Jaderberg et al., 2014), and quantization (Han et al., 2016), to name a few. Several model compression techniques factorize trained filters to reduce the computation complexity (Zhang et al., 2015; Li et al., 2019a). Our proposed block structure also benefit from the reduced computational complexity too by factorizing convolution filters. However, most previous compression techniques have been suggested as post-processing steps that are applied after initial training. Therefore, their accuracy is usually bounded by the approximated base models. In contrast, our primary goal in factorizing convolution operators is to find a common filter basis shared by iterative convolution layers, so the weights of the shared filter basis are learned from scratch, not from decomposing pretrained filters. Hence, unlike previous compression techniques, the performance of our approach is not limited by the original models, and the experimental results show that our parameter-sharing approach consistently outperforms the overparameterized counterpart networks on various datasets and tasks.

## 3 Proposed Method

In this section, we describe the design of recursive convolution blocks and discuss how to train them to deal with the vanishing/exploding gradients problem.

### 3.1 Recursive Sharing of a Filter Basis

Naive sharing of convolution filters across repetitive layers degrades the overall performance due to limited representation power of individual layers. We argue that convolution filters can be separated into fundamentally shareable components and non-shareable components, and the performance degradation can be prevented by sharing only fundamentally shareable parts.

More formally, we consider a convolution layer, shown in Figure 1 (a), that has $S$ input channels, $T$ output channels, and a set of filters $W = \{W_t \in R^{k \times k \times S}, t \in [1..T]\}$. Each filter $W_t$ can be decomposed into a filter basis $W_{basis}$ and coefficients $\alpha$:

$$W_t = \sum_{r=1}^{R} \alpha_t^r W_{basis}^r, \tag{1}$$

where $W_{basis} = \{W_{basis}^r \in \mathbb{R}^{k \times k \times S}, r \in [1..R]\}$ is a filter basis, and $\alpha = \{\alpha_t^r \in \mathbb{R}, r \in [1..R], t \in [1..T]\}$ is scalar coefficients. In Equation 1, $R$ is the rank of the basis. In a typical convolution layer, output feature maps $V_t \in \mathbb{R}^{w \times h \times T}, t \in [1..T]$ are obtained by the convolution between input feature maps $U \in \mathbb{R}^{w \times h \times S}$ and filters $W_t, t \in [1..T]$. With Equation 1, this convolution can be rewritten as

follows:

$$V_t = U * W_t = U * \sum_{r=1}^{R} \alpha_t^r W_{basis}^r = \sum_{r=1}^{R} \alpha_t^r (U * W_{basis}^r), \text{ where } t \in [1..T]. \tag{2}$$

In Equation 2, the order of the convolution operation and the linear combination of filter basis is reordered according to the linearity of convolution operators. Therefore, a conventional convolution layer can be replaced with two successive convolution layers as shown in Figure 1-(b).

This factorized convolution filters suggest a recursive architecture, shown in Figure 1-(c), in which a common filter basis is used for iterative convolution layers. Since all filters in the recursive loop have a common filter basis, they are positioned in the same vector subspace. However, their representation capability is still retained since their coordinates in the subspace are diversified by their respective non-shared coefficients.

## 3.2 Orthonormality of Shared Filter Bases

For recursive sharing of a filter basis, as in Figure 1-(c), we need to find an optimal filter basis that can be used by iterative convolution layers without loss of performance. Although this optimal filter basis can be searched by typical gradient-based optimization such as SGD, one major problem is that the exploding/vanishing gradients problem of recursively shared filter bases can prevent proper search of optimization space (Pascanu et al., 2013).

More formally, we consider a series of $N$ factorized convolution blocks, in which a filter basis $W_{basis}$ is shared $N$ times, as in Figure 1-(c). Let $\mathbf{x}^i$ be the input of the $i$-th convolution block, and $a^{i+1}$ be the output of the convolution of $\mathbf{x}^i$ with the filter basis $W_{basis}$

$$a^i(\mathbf{x}^{i-1}) = W_{basis}^\top \mathbf{x}^{i-1}. \tag{3}$$

In Equation 3, $W_{basis} \in \mathbb{R}^{k^2 S \times R}$ is a reshaped filter basis that has basis elements at its columns. We assume that input $\mathbf{x}$ is properly adapted (e.g., with *im2col* operations) to express convolutions using a matrix-matrix multiplication. Since $W_{basis}$ is shared by $N$ recursive convolution blocks, the gradient of $W_{basis}$ for some loss function $L$ is:

$$\frac{\partial L}{\partial W_{basis}} = \sum_{i=1}^{N} \frac{\partial L}{\partial a^N} \prod_{j=i}^{N-1} \left( \frac{\partial a^{j+1}}{\partial a^j} \right) \frac{\partial a^i}{\partial W_{basis}}, \tag{4}$$

, where

$$\frac{\partial a^{j+1}}{\partial a^j} = \frac{\partial a^{j+1}}{\partial \mathbf{x}^j} \frac{\partial \mathbf{x}^j}{\partial a^j} = W_{basis} \frac{\partial \mathbf{x}^j}{\partial a^j} \tag{5}$$

If we plug $W_{basis} \frac{\partial \mathbf{x}^j}{\partial a^j}$ in Equation 5 into Equation 4, we can see that $\prod \frac{\partial a^{j+1}}{\partial a^j}$ is the term that makes gradients unstable since $W_{basis}$ is multiplied many times. This exploding/vanishing gradients can be controlled to a large extent by keeping $W_{basis}$ close to orthogonal (Vorontsov et al., 2017). For instance, if $W_{basis}$ admits eigendecomposition, $[W_{basis}]^N$ can be rewritten as follows:

$$[W_{basis}]^N = [Q \Lambda Q^{-1}]^N = Q \Lambda^N Q^{-1}, \tag{6}$$

where $\Lambda$ is a diagonal matrix with the eigenvalues placed on the diagonal and $Q$ is a matrix composed of the corresponding eigenvectors. If $W_{basis}$ is orthogonal, $[W_{basis}]^N$ neither explodes nor vanishes, since all the eigenvalues of an orthogonal matrix have absolute value of 1. This result shows that a shared filter basis with orthonormal elements ensures that forward and backward signals neither explode nor vanish.

Based on this result, we propose the *orthogonality regularization* [2] to enforce orthonormality to shared filter bases during training. For instance, when convolution operators in each residual block group of a ResNet shares a filter basis, the objective function $L_R$ can be defined to have the orthogonality regularization term in addition to the original loss $L$:

$$L_R = L + \lambda \sum_{g}^{G} \| W_{basis}^{(g)}{}^\top \cdot W_{basis}^{(g)} - I \|^2, \tag{7}$$

---

[2] Technically, the columns (or rows) of an orthogonal matrix form an *orthonormal basis*. However, we use the term *orthogonality regularization* since orthonormal bases have no proper term for their matrix-form.

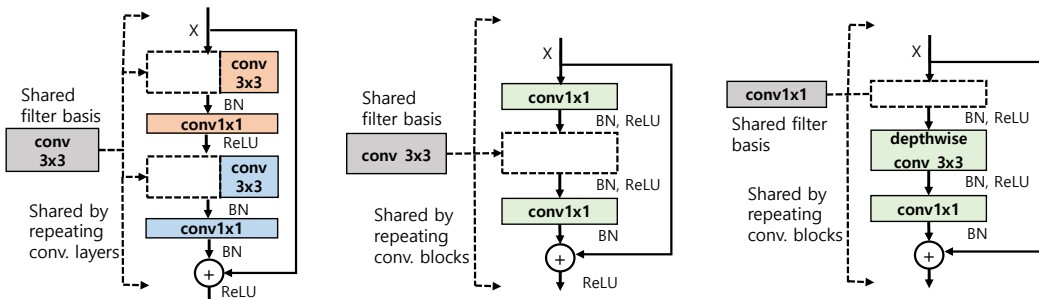

(a) Shared block for ResNet34     (b) Shared block for ResNet50     (c) Shared block for MobileNetV2

Figure 2: The structure of residual blocks with a shared filter basis. For ResNet34, two full convolution operators are factorized to uncover a shared filter basis. For ResNets with bottleneck blocks (ResNet50/101) and MobileNetV2, one or two computationally expensive convolution operators of each convolution block are selected as shared filter bases and the other convolution operators are designated as non-shared coefficients. In (a), the filter basis has non-shared elements to further increase representation capability.

where $W_{basis}^{(g)}$ is a shared filter basis for $g$-th residual block group and $\lambda$ is a hyperparameter.

In Equations 4 and 5, we also need to ensure that the norm of $\frac{\partial \mathbf{x}^j}{\partial a^j}$ is bounded for stability during forward and backward passes (Pascanu et al., 2013). It is shown that batch normalization after non-linear activation at each convolution layer ensures healthy norms (Ioffe & Szegedy, 2015; Guo et al., 2019; Jastrzebski et al., 2018). In Section 4.4, we empirically show that the proposed orthogonality regularization and batch normalization similarly improve the flow of gradients during training.

### 3.3 Enhancing Representation Capability

When convolution operators share a common filter basis, they are all in the same vector subspace. Therefore, if the rank of the filter basis is too low, it might limit the representation capability of the convolution operators sharing the filter basis. Conversely, if the rank of the shared filter basis is too high (e.g., $R \geq T$), the computational gain of factorized structure is mitigated. One way to increase the representational power of each convolution operator, while still maintaining its computational complexity low, is placing the convolution operators in different subspace by adding a small number of non-shared elements to the filter basis. For instance, we build a filter basis $W_{basis}$ by combining shared elements and non-shared elements:

$$W_{basis} = W_{bs\_shared} \cup W_{bs\_unique}, \tag{8}$$

where $W_{bs\_shared} = \{W_{bs\_shared}^r \in \mathbb{R}^{k \times k \times S}, r \in [1..n]\}$ are shared filter basis elements, and $W_{bs\_unique} = \{W_{bs\_unique}^r \in \mathbb{R}^{k \times k \times S}, r \in [n+1..R]\}$ are non-shared filter basis elements. For example, Figure 2-(a) shows that the filter basis of ResNet34 is compose of both shared and non-shared elements. One disadvantage of this hybrid scheme is that non-shared filter basis elements require more parameters. The ratio of non-shared basis elements can be varied to control the tradeoffs. But, our results in Section A.1 show that providing only a few non-shared elements to a filter basis is enough to achieve high performance.

## 4 Experiments

In this section, we perform a series of experiments on image classification and object detection using several modern networks as base models. We also analyze the effect of the orthogonality regularization.

For ResNets with conventional convolution filters (e.g, ResNet34), a filter is replaced by the proposed factorized convolution block as shown in Figure 2-(a). A filter basis is shared within a residual block group having same kernel dimensions. Throughout the experiments, we denote by ResNet$L$-S$s$U$u$ a ResNet with $L$ layers that has a filter basis with a $s$-to-$u$ ratio of shared elements and non-shared elements.

ResNets with bottleneck blocks (e.g., ResNet50) and MobileNetV2 already have decomposed block structures. Therefore, we designate one or two convolution operators with the largest parameters in each block as filter bases sharing weights across iterative blocks, as shown in Figure 2-(b) and -(c). During the training, the proposed orthogonality regularization is applied to enforce the elements of the shared filter bases orthonormal. In these models, we leave at least one convolution operator as non-shared coefficients to improve expression ability. In Section 4.1, we explore the effect of sharing both a filter basis and coefficients in residual groups.

During experiments, all programs for training and evaluation run on PCs equipped either with four RTX 2080Ti GPUs or two RTX 3090 GPUs and an Intel i9-10900X CPU @3.7GHz.

## 4.1 Image Classification on ImageNet

We evaluate our method on the ILSVRC2012 dataset (Russakovsky et al., 2015) that has 1000 classes. The dataset consists of 1.28M training and 50K validation images. We use ResNets and MobileNetV2 as base models. We train the ResNet-derived models for 150 epochs with SGD optimizer with a mini-batch size of 256, a weight decay of 1e-4, and a momentum of 0.9. The learning rate starts with 0.1 and decays by 0.1 at 60-th, 100-th, and 140-th epochs. MobileNetV2 and our MobileNetV2-Shared models are trained for 300 epochs with a weight decay of 1e-5. Its learning rate starts with 0.1 and decays by 0.1 at 150-th, 225-th, and 285-th epochs.

Table 1: Error (%) on ImageNet. In ResNet50/101-Shared[‡] and MobileNetV2-Shared[‡], first two convolution operators of each residual block are designated as recursively shared filter bases. In ResNet50-Shared-All, both a filter basis and coefficients are shared recursively across blocks of each residual group. In ResNet50-Shared-NoOrthoReg, orthogonality regulaization is not applied to ResNet50-Shared during training. Latency is measured on the Nvidia Jetson TX2 embedded board (GPU, batch size = 1).

| Baseline | Model | Params | FLOPs | top-1 | top-5 | Latency |
|---|---|---|---|---|---|---|
| ResNet34 | ResNet34 (baseline) | 21.80M | 3.67G | 26.70 | 8.58 | 33.6ms |
| | Filter Pruning (Li et al., 2017) | 19.30M | 2.76G | 27.83 | - | - |
| | ResNet34-S48U1 (ours) | 11.79M | 3.26G | **26.67** | **8.54** | 38.6ms |
| ResNet50 | ResNet50 (baseline) | 25.56M | 4.11G | 23.85 | 7.13 | 43.8ms |
| | Versatile-ResNet50 (Wang et al., 2018) | 19.0M | 3.2G | 24.5 | 7.6 | - |
| | FSNet (Yang et al., 2020) | 13.9M | - | 26.89 | 8.63 | - |
| | ResNet50-Shared (ours) | 20.51M | 4.11G | **23.64** | **6.98** | 43.3ms |
| | ResNet50-Shared[‡] (ours) | 18.26M | 4.11G | 23.95 | 7.14 | 43.3ms |
| | ResNet50-Shared-All | 16.02M | 4.11G | 24.35 | 7.41 | - |
| | ResNet50-Shared-NoOrthoReg | 20.51M | 4.11G | 24.19 | 7.34 | - |
| ResNet101 | ResNet101 (baseline) | 44.55M | 7.83G | 22.63 | 6.44 | 73.2ms |
| | ResNet101-Shared (ours) | 29.47M | 7.83G | **22.31** | 6.47 | 72.9ms |
| MobileNetV2 | MobileNetV2 (baseline) | 3.50M | 0.33G | 28.0 | 9.71 | 18.4ms |
| | DR-MobileNetV2 (Guo et al., 2019) | 2.96M | 0.27G | 28.2 | 9.72 | - |
| | MobileNetV2-Shared (ours) | 3.24M | 0.33G | **27.61** | **9.34** | 17.9ms |
| | MobileNetV2-Shared[‡] (ours) | 2.98M | 0.33G | 28.21 | 9.85 | 17.8ms |

In Table 1, we compare our results with competing techniques. Most parameter-sharing and compression techniques save parameters at the cost of performance. Unlikely, the results in Table 1 show that our approach consistently outperforms the counterpart networks while saving a significant amount of parameters. For example, ResNet34-S48U1 outperforms the counterpart ResNet34 while only using 54.0% parameters. Since our models derived from ResNet50/101 and MobileNetV2 already have factorized convolution blocks, their overall parameter-saving is not as high as ResNet34-S48U1's. However, they still save about 19.8%, 33.9% and 7.5% parameters, respectively, while outperforming the baselines. The benefit of our method is more pronounced in deeper networks. In ResNet101, for example, a filter basis is shared by up to 23 recursive convolution layers, resulting in 35.7% reduction of parameters. In Table 1, ResNet50-Shared-NoOrhtoReg shows the effect of not applying the orthogonality regularization on ResNet50-Shared. Our result shows that its top1 accuracy drops to 75.81%, that is 0.34% lower than the counterpart ResNet50. In contrast, when the orthogonality regularization is applied, the same model achieves 76.35%, which is 0.21% higher than

the counterpart ResNet50. This 0.55% improvement is obtained simply by applying orthogonality regularization. In Table 1, ResNet50-Shared-All shows the effect of sharing not only a filter basis but also coefficients recursively within residual groups. The result shows that we can save further 18% parameters (reduced from 20.51M to 16.02M) of ResNet50-Shared, but its top-1 (top-5) accuracy drops to 75.65% (92.59%), which is 0.5% lower than the counterpart ResNet50. This result shows that having unshared coefficients is important for high performance. Yang et al. (2019) also demonstrates that sharing coefficients drops the accuracy by additional 2%.

Although ResNet34-S48U1 requires lower FLOPs than the counterpart ResNet34, it takes 14% longer latency on Jetson TX2. This overhead mostly comes from separated convolution operations for shared and non-shared filter basis elements. Current GPU-based deep learning libraries are not optimized to process such separated operations efficiently. Unlikely, ResNet50/101-Shared and MobileNetV2-Shared do not have such separated convolution operations, and, hence, their latency is slightly lower on the device that is constrained by limited cache and memory.

## 4.2 Image Classification on CIFAR

We evaluate the effectiveness of our method on various modern CNNs with the CIFAR dataset that has 50,000 and 10,000 $32 \times 32$ images for training and testing, respectively. For training networks, we follow a similar training scheme in He et al. (2016). Standardized data-augmentation and normalization are applied to input data. Networks are trained for 300 epochs with SGD optimizer with a weight decay of 5e-4 and a momentum of 0.9. The learning rate is initialized to 0.1 and is decayed by 0.1 at 50% and 75% of the epochs.

Table 2: Error (%) on CIFAR-100. '*' denotes that the orthogonality regularization is not applied.

| Baseline | Model | Params | FLOPs | Error |
|---|---|---|---|---|
| ResNet34 | ResNet34 (baseline) | 21.33M | 1.17G | 22.49 |
| | ResNet34-S32U1* (ours) | 7.73M | 0.78G | 22.92 |
| | ResNet34-S32U1 (ours) | 7.73M | 0.78G | **21.84** |
| DenseNet121 | DenseNet121 (baseline) | 7.05M | 0.91G | 21.95 |
| | DenseNet121-S16U1 (ours) | 5.08M | 0.72G | 22.15 |
| ResNeXt50 | ResNeXt50 (baseline) | 23.17M | 1.36G | 20.71 |
| | ResNeXt50-S64U4 (ours) | 19.3M | 1.19G | **20.09** |

Table 3: Error (%) on CIFAR-10. '‡' denotes having 2 shared bases in each convolution block group. '⋆' denotes that the orthogonality regularization is not applied.

| Baseline | Model | Params | FLOPs | Error |
|---|---|---|---|---|
| ResNet32 | ResNet32 (baseline) | 0.46M | 0.07G | 7.51 |
| | ResNet32-S16U1‡ (ours) | 0.24M | 0.08G | **6.95** |
| ResNet56 | ResNet56 (baseline) | 0.85M | 0.16G | 6.97 |
| | Filter Pruning (Li et al., 2017) | 0.77M | 0.09G | 6.94 |
| | KSE (Li et al., 2019b) | 0.43M | 0.06G | 6.77 |
| | DR-Res 40 (Guo et al., 2019) | 0.50M | 0.11G | 6.51 |
| | ResNet56-S16U1⋆ (ours) | 0.27M | 0.15G | 7.70 |
| | ResNet56-S16U1 (ours) | 0.27M | 0.15G | 7.46 |
| | ResNet56-S16U1‡ (ours) | 0.31M | 0.15G | **6.33** |

Table 2 shows the results on CIFAR-100. Networks trained with the proposed method consistently outperform their counterparts. For instance, ResNet34-S32U1 requires only 36.2% parameters and 66.6% FLOPs of the counterpart ResNet34 while achieving lower test error (21.84%) than much deeper ResNet50 (22.36%). To show the generality of our work, we apply the proposed method to DenseNet (Huang et al., 2017), and ResNeXt (Xie et al., 2017). Although the overall gain is not as pronounced as ResNets', we still observe reduction of resource usages in these networks. For instance, ResNeXt50-S64U4 outperforms the counterpart ResNeXt50 while saving parameters and FLOPs by 16.7% and 12.1%, respectively. In ResNeXt, the gain is limited since they mainly exploit group convolutions; each group convolution is decomposed for filter basis sharing in our network.

The result on CIFAR-10 with ResNets is presented in Table 3. Unlike networks on CIFAR-100, networks on CIFAR-10 has much fewer channels (e.g. 16 channels in the first residual block group) and, hence, projecting filters to such low dimensional subspace might limit the performance of the networks. For instance, in ResNet32-S8U1, filters are supposed to be projected onto 9 dimensional subspace consisting of 8 shared and 1 non-shared filter basis elements. Further, for deeper networks such as ResNet56, a filter basis is supposed to be shared by many residual blocks in the group, and it can damage the performance. For example, every filter basis in ResNet56-S16U1 is shared by 8 residual blocks, or 16 convolution operators. Due to this excessive sharing, though ResNet56-S16U1 saves 41.3% parameters, its testing error (7.46%) is higher than the counterpart ResNet56's (6.97%).

To remedy this problem, we introduce a variant, in which each residual block group of the networks uses 2 shared bases; one basis is shared by the first convolution operators of recursive residual blocks, and the other basis is shared by the second convolution operators. In Table 3, networks with a '‡' mark denote this variant. Though this variant slightly increases the parameters of the networks, it can prevent excessive sharing of parameters. For example, although ResNet56-S16U1‡ needs 0.04M more parameters for additional shared bases, it still saves 63% parameters of the counterpart ResNet56 and achieves lower testing error of 6.33%.

In Table 3, we compare our results with similar state-of-the-art techniques. Our method achieves better performance and parameter-saving than other approaches such as filter pruning (Li et al., 2017), kernel clustering (Li et al., 2019b), and recursive parameter sharing (Guo et al., 2019).

## 4.3 Object Detection on MS COCO

In order to explore the generalization ability of our approach, we next use COCO 2017 dataset on object detection task using Faster R-CNN (Ren et al., 2017), Mask R-CNN (He et al., 2017a), and RetinaNet (Lin et al., 2017) as detectors. We compare ResNet50/101 and our ResNet50/101-Shared as backbone networks of the detectors. These backbone networks are pre-trained on ImageNet, then are transferred to MS COCO by fine-tuning. We use MMDetection (Chen et al., 2019) toolbox and employ default settings for training and evaluation. All networks are trained on `train2017` for 12 epochs using SGD with weight decay of 1e-4, momentum of 0.9 and mini-batch size of 8 (2 examples per GPU). The learning rate is initialized to 0.01 and decays by 0.1 at 8-th and 11-th epochs.

Table 4: Object detection results on COCO 2017 validation set. ResNet50-Shared‡ uses first two convolution operators of each residual block as recursively shared filter bases.

| Backbone | Detector | #Params | GFLOPs | $AP$ | $AP_{50}$ | $AP_{75}$ | $AP_S$ | $AP_M$ | $AP_L$ |
|---|---|---|---|---|---|---|---|---|---|
| ResNet50 (baseline) | | 41.53M | 207.07 | 36.4 | 58.2 | 39.2 | 21.8 | 40.0 | 46.2 |
| -Shared (ours) | Faster | **36.70**M | 206.87 | **37.2** | 58.1 | 40.2 | 21.6 | 40.8 | 47.9 |
| -Shared‡ (ours) | R-CNN | **34.46**M | 206.87 | **36.6** | 57.4 | 39.8 | 21.4 | 40.1 | 47.3 |
| ResNet101 (baseline) | | 60.52M | 283.14 | 38.7 | 60.6 | 41.9 | 22.7 | 43.2 | 50.4 |
| -Shared (ours) | | **45.67**M | 282.84 | **39.0** | 59.7 | 42.7 | 22.4 | 42.7 | 50.8 |
| ResNet50 (baseline) | | 44.18M | 275.58 | 37.2 | 58.9 | 40.3 | 22.2 | 40.7 | 48.0 |
| -Shared (ours) | Mask | **39.35**M | 259.94 | **37.9** | 58.4 | 41.4 | 22.4 | 41.2 | 49.2 |
| -Shared‡ (ours) | R-CNN | **37.10**M | 259.94 | **37.3** | 57.6 | 40.7 | 21.5 | 40.3 | 48.5 |
| ResNet101 (baseline) | | 67.17M | 351.65 | 39.4 | 60.9 | 43.3 | 23.0 | 43.7 | 51.4 |
| -Shared (ours) | | **48.31**M | 335.91 | **39.8** | 60.3 | 43.6 | 22.7 | 43.5 | 51.9 |
| ResNet50 (baseline) | | 37.74M | 239.32 | 35.6 | 55.5 | 38.2 | 20.0 | 39.6 | 46.5 |
| -Shared (ours) | | **32.92**M | 239.12 | **36.2** | 55.0 | 38.6 | 20.3 | 39.7 | 47.1 |
| -Shared‡ (ours) | RetinaNet | **30.67**M | 239.12 | 35.6 | 54.1 | 38.2 | 19.8 | 39.2 | 46.9 |
| ResNet101 (baseline) | | 56.74M | 315.39 | 37.7 | 57.5 | 40.4 | 21.1 | 42.2 | 49.5 |
| -Shared (ours) | | **41.88**M | 315.09 | 37.7 | 56.8 | 40.3 | 21.2 | 41.5 | 49.5 |

Table 5: Instance segmentation results using Mask R-CNN on COCO val2017.

| Backbone | $AP$ | $AP_{50}$ | $AP_{75}$ | $AP_S$ | $AP_M$ | $AP_L$ |
|---|---|---|---|---|---|---|
| ResNet50 (baseline) | 34.1 | 55.5 | 36.2 | 16.1 | 36.7 | 50.0 |
| -Shared (ours) | **34.5** | 55.4 | 36.9 | 18.9 | 37.7 | 46.5 |
| ResNet101 (baseline) | 35.9 | 57.7 | 38.4 | 16.8 | 39.1 | 53.6 |
| -Shared (ours) | 35.9 | 57.2 | 38.3 | 19.1 | 39.2 | 48.6 |

Table 4 shows the results on `val2017` containing 5000 images. Our backbone networks with shared filter bases consistently outperform baselines in all detectors in terms of COCO's standard metric AP while saving up to 28.1% parameters. In Table 5, our models also achieve similar performance improvement in instance segmentation using Mask R-CNN. This consistent performance improvement is obtained simply by replacing the backbone networks with ours and it demonstrates that the proposed recursive convolution block design and the orthogonality regularization enables learning better feature representations with a smaller amount of parameters.

## 4.4 Analysis: Effect of Orthogonality Regularization

To investigate the effect of the orthogonality regularization during training, we track the flows of gradients while training ResNet34-S16U1. Figure 3 shows the traces of the maximum and the mean gradients flowing in the filter bases during an epoch. On every 10 iterations of a batch, the maximum and the mean gradients are overlapped on top of the old plots. Therefore, the bars look darker if bars are more overlapped. Jastrzebski et al. (2018) and Guo et al. (2019) showed that unshared batch normalization (BN) strategy mitigates the vanishing/exploding gradients problem, and our result in Figure 3-(b) shows that unshared BNs following shared filter bases improve the flow of gradients. When the proposed orthogonality regularization is applied to the shared filter bases, in Figure 3-(c), similar effect is observed on gradients. When both unshared BNs and the orthogonality regularization are applied together, in Figure 3-(d), further stronger, but still bounded, flows of gradients are observed. This trend is consistently observed during training. We conjecture that this healthy flow of gradients enables learning better feature representations during optimization process, resulting in superior performance on various tasks.

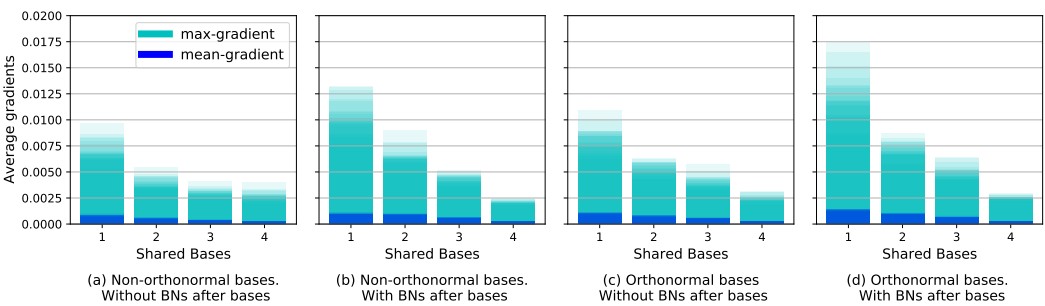

Figure 3: The flows of gradients in 4 shared filter bases of ResNet34-S16U1 at the same epoch. For comparison, the orthogonality regularization and the batch normalization (BN) following the filter bases are turned on and off. In (b) and (c), BNs and the orthogonality regularization, respectively, improve the flow of gradients. In (d), when both BNs and orthogonality regularization are applied together, the strongest flow of gradients is observed. This trend is consistently observed during the training.

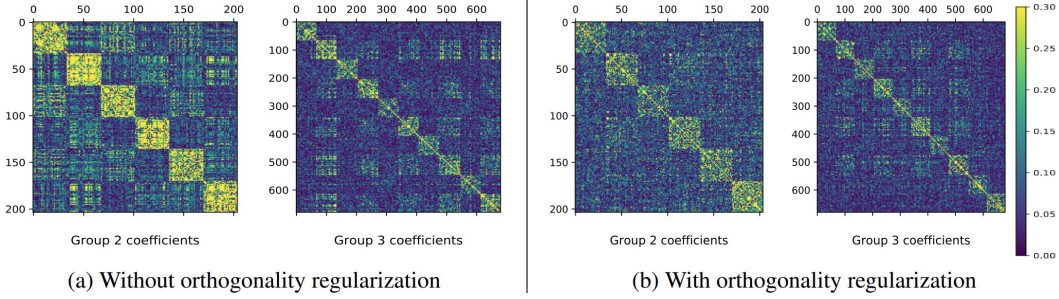

Figure 4: Cosine similarities of coefficients of the 2nd and the 3rd residual block groups in ResNet34-S16U1. X and Y axes are indexes to the coefficients of the residual block groups sharing filter bases. Brighter colors indicate higher similarity. In (b), when the orthogonality regularization is applied, the similarity is clearly lowered, implying less redundancy in parameters.

To further analyze the effect of the orthogonality regularization, in Figure 4, we illustrate absolute cosine similarities of all coefficients of the 2nd and the 3rd residual block groups of ResNet34-S16U1. The X and Y axes display the indexes to the coefficients in the residual blocks. In Figure 4-(b), we can clearly see that coefficients manifest lower similarities when the orthogonality regularization is applied. In Figure 4-(a), when the orthogonality regularization is not applied, an interesting grid pattern is observed in Figure 4-(a). This repetitive grid pattern might be related to ResNets' nature of iterative feature refinement (Jastrzebski et al., 2018). However, such high cosine similarity is directly related to the higher redundancy in the networks. When our orthogonality regularization is applied, such repetitive patterns are less evident, implying that the recursive convolution layers learn better feature representations with less redundancy.

## 5 Conclusions

We introduce a recursive convolution block design and effective training method for parameter-efficient CNNs. In this work, a common filter basis, shared by repeating convolution layers, is learned while effectively avoiding the vanishing/exploding gradients problem through the proposed orthogonality regularization. Experimental results on image classification and image detection show that our approach consistently outperforms over-parameterized counterpart models while significantly saving parameters. This consistent performance improvement demonstrates that the proposed approach enables effective learning of better feature representations while a significant amount of parameters are shared. We believe that the proposed recursive convolution blocks and training method suggests important possibilities for neural architecture search (NAS) to explore resource-efficient CNNs.

## Acknowledgments and Disclosure of Funding

We thank the anonymous reviewers for their constructive comments and suggestions. This work was supported by the National Research Foundation of Korea (NRF) Grants Funded by the Ministry of Science and ICT under Grant NRF-2019R1F1A1060959.

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
