# A Appendix

## A.1 Effects of Ranks of Shared/Unshared Bases

Figure 5 shows test errors on CIFAR-100 as parameters and FLOPs are increased by varying the number of shared/non-shared filter basis elements of networks. In general, the higher performance is expected with the more parameters. We observe that this presumption is true for shared basis elements. For instance, when the number of shared basis elements $s$ is varied from 8 to 32, the test error sharply decreases from 23.1% to 21.7%. However, non-shared basis elements manifest counter-intuitive results. Although a small number of non-shared basis elements (e.g., $u = 1$) are clearly beneficial to the performance, the higher $u$'s do not always lead to the higher performance. For instance, when $u = 4$, both ResNet34-S16U$u$ and ResNet34-S32U$u$ show the worst performance. This result demonstrates the difficulty of training networks with larger parameters. Further study is required for this problem.

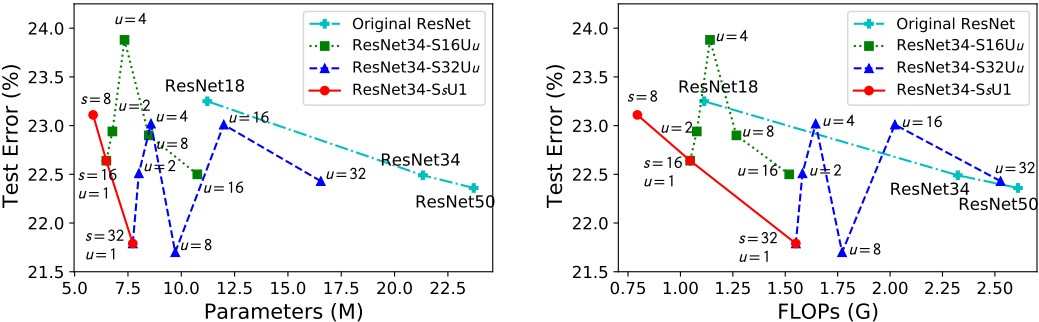

Figure 5: Testing errors vs. the number of parameters and FLOPs on CIFAR-100. The number of shared basis elements ($s$), and non-shared basis elements ($u$) are varied. Using more shared basis elements results in better performance. In contrast, using more non-shared elements does not always improve performance, implying the difficulty of training networks with larger parameters.