# OpenReview forum: "Deeply Shared Filter Bases for  Parameter-Efficient  Convolutional Neural Networks"
_NeurIPS.cc/2021/Conference — NeurIPS 2021 Poster_

### Official Review · Reviewer_k6be · 2021-07-12

**Rating:** 6
**Confidence:** 5

**Summary:**

This paper proposes a recursive convolutional block design for compact CNNs with efficient training. The convolutional filters (across all layers) are decomposed into shared basis filters and non-shared expanding coefficients, both of which are trained in an end-to-end manner. To avoid vanishing/exploding gradients, orthogonality regularization is imposed, and the efficacy of which is shown through both theoretical analysis and empirical verification. The proposed architecture consistently outperforms the corresponding baseline in terms of accuracy with a reduced model size.

**Limitations And Societal Impact:**

No comment.

**Main Review:**

** Pros **
1. The paper is very well-written and well-organized. It is easy to read.
2. Imposing orthogonality constraint on the shared basis is interesting, and its effectiveness is explained both theoretically and empirically.
3. A small portion of non-shared layer-specific basis is also used to maintain expressiveness of the model.
4. Extensive numerical experiments have been conducted to demonstrate the effectiveness of the model.

** Cons **
1. Missing citation: the idea of filter decomposition into shared basis and non-shared coefficients is not new, e.g., it is also proposed in Qiu et al., "DCFNet: Deep Neural Network with Decomposed Convolutional Filters", ICML 2018.
2. Figure 3 reports the effectiveness of regularization in preventing vanishing gradients. Is there a visual illustration demonstrating how the regularization can prevent exploding gradients?
3. The choice of lambda is rarely mentioned in the manuscript. An ablation study on lambda should be included in section 4.4.

Overall it is an interesting paper (even though the idea of filter decomposition has been widely studied and adopted), but I would love to see the feedbacks from the authors.

** Review after author feedback **
The authors have partially addressed my concerns. I am raising my rating to 6.

**Time Spent Reviewing:**

3

---

> ### Author Response · Authors · 2021-08-10
> **Thank you for your comment**
>
> 1. We appreciate the reviewers for recommending more related papers. We will add discussion about [1]-[4] in the revision. Although the approaches in [1]~[4] are different in details, they are similar to our work since they all share parameters and combine them to build  layer-specific filters. However, our work is the only work, at least among [1]-[4], that demonstrates better performance than counterpart networks on large datasets (ImageNet and COCO). We further demonstrate the effectiveness of our approach using ResNet101. As far as we know, our approach is the only work, at least among [1]-[4],  demonstrating effective parameter sharing in ‘deep’ networks such as ResNet101 on ImageNet and COCO. In our approach, a filter basis is shared by up to 22 successive convolution blocks of ResNet101 while even outperforming counterpart ResNet101 by 0.32% in accuracy (top1).
> We believe that this performance improvement comes from gradient issues of shared parameters.
>
> 2. Fig 3 shows both the ‘maximum’ and the ‘mean’ gradients during training. Since all approaches have bounded maximum gradients, their gradients do not explode. Since modern networks already have several mechanisms to improve the flow of gradients, e.g., skip connections and batch normalization,
> we cannot claim that gradient explosions are prevented only because of orthogonality regularization. The bounded gradients in Fig. 3 are the result of these mechanisms and our orthogonality regularization combined.
>
> 3. “lambda” is a hyperparameter that determines how orthogonality regularization affects the loss. During evaluation, we have tested lambdas between 0.1 and 1000 and found that our approach is not very sensitive to the value of lambdas and obtained best results when lambda is between 10 and 100. Hence, we use lambda=10 as default during all experiments. We will mention this explicitly in the revision.
>
> [1]  Qiu et al. "DCFNet: Deep Neural Network with Decomposed Convolutional Filters", ICML 2018
>
> [2] Yang et al. LegoNet: Efficient Convolutional Neural Networks with Lego Filters. ICML 2019.
>
> [3] Bhalgat et al. Structured Convolutions for Efficient Neural Network Design. NeurIPS 2020.
>
> [4] Ha et al. HyperNetworks, ICLR 2017

---

### Official Review · Reviewer_oyjZ · 2021-07-16

**Rating:** 6
**Confidence:** 4

**Summary:**

This paper describes a method to share parameters between iterations of a residual block.  Rather than sharing all parameters or none, a filter basis W_basis and mixture coefficients alpha are defined; the basis is shared between layers, while the coefficients are unshared.  A theoretical argument shows a potential cause of training instability due to repeated application of a shared weights term; an orthonormality regularizer is applied to the weights basis, to help alleviate vanishing/exploding gradients.  Measurements are performed on ImageNet, CIFAR and COCO, applied to a variety of resnets and mobilenet.  The resulting model uses about half as many parameters (though similar number of FLOPs) as a baseline model, and achieves similar or slightly better accuracy.


**Limitations And Societal Impact:**

yes

**Main Review:**

The system is similar to Savarese and Maire 2019, which is described in the related work.  The differences here appear to be (a) the analysis of repeated weights that may cause large/small gradients, (b) the orthonormality regularizer, and (c) the use of small numbers of unshared weights in addition to basis-generated weights.

While the theoretical argument seemed persuading, and the final system in showed good results in the experiments, I think additional experimental baselines and comparisons would make the benefits of the method more convincing.  In particular, there are no comparisons for using no unshared elements (U=0), or the naive approach of fully sharing weights within residual groups.  (Savarese and Maire is also not compared directly, though a configuration of this system without regularizer may be equivalent).

Likewise, the effect of the orthogonality regularizer is demonstrated to create larger gradients (if I'm interpreting the plot correctly, see below additional comments), but the final effect on CIFAR-100 is about 1% accuracy gain in the best case, which is good but seems a little small compared to the motivation of unstable gradients --- might there be other effects, e.g. on the number of steps needed in the training schedule?

Overall, the system improves upon similar parameter sharing schemes, with good results.  But it's unclear how much it helps relative to a few other simple baselines, described above.  Including these would strengthen the experimental case.



Additional comments and questions:


- Steerable weights in related work:  There is some work on steerable cnns that I feel is related and might be discussed; these systems use transformed weights for enforced equivariance, e.g. https://arxiv.org/pdf/1612.08498.pdf, https://openaccess.thecvf.com/content_cvpr_2018/CameraReady/3214.pdf


- What is the effect of orthgonality regularizer when training on ImageNet?  The difference in classification error from this term is only shown for a subset of the cifar experiments (resnet34 on cifar100 and resent56 on cifar10).  What happens if orthogonality regularizer is not used on ImageNet?  More comprehensive experiments on this would be more convincing.

- What is the effect of lambda in the loss?  To me it looks it would be relatively unsensitive to the exact value, and it should be set fairly low, around 0.01 or lower.  But would be good to confirm by showing an experiment for this.


- Compare to naively shared weights (where successive block iters use the same weights):  I didn't see this comparison in the paper --- in addition to unshared baseline, it would be good to see param count and accuracy drop of the baseline model with weights shared in each stage, to confirm its accuracy loss.


- Effect of U0 vs U1:  The text says that a small number of unshared weights help performacne, but I didn't see this evaluated; the smallest number of unshared basis elements is 1.  What about 0 (all shared)?



- sec 4.4 Fig 3:  I don't understand how to interpret this plot, more explanation may be needed.  What does the varying lightness of the light blue bars mean --- are there multiple overlapping translucent bars, and why?  Also, what does a value of 0.0075 mean --- relative to what?  Are these per-element gradients, and if so, what is this value relative to the weight, and why is this value good?  What about the step size, incorporating the optimizer learning rate, momentum etc?  If gradients are larger or better conditioned, can the training schedule be reduced?


- sec 4.4 Fig 4:  What are the similarity matrices for groups 1 and 4?  Showing only 2 and 3 seems incomplete to me.  Do they show the same pattern?  And if not, what might be a cause for difference?


- eq. 5 and text:  Since the alpha coefficients don't appear here explicitly, and are part of the repeated net (with unshared weights), I think it could be useful to explain where they are here (as part of d x^j / d a^j), and relate to the paragraph on l.156 on what influences their stability


- Appendix fig A.1:  u=0 is missing from these curves, so it's hard to conclude that having one unshared element helps, as the text says.  Also would be good to have error bars over a few different random inits, as the errors are jumping around 1% out of 22%.  This is a nice plot for the supplement, though, I thought it was interesting to see.



- l.148 "If Wbasis is orthogonal ... ":  this sentence should probably also use "orthonormal"

- l.227: resnet50 does not appear in the table







**Time Spent Reviewing:**

3

---

> ### Author Response · Authors · 2021-08-10
> **Thank you for your comment**
>
> 1. “I think additional experimental baselines and comparisons would make the benefits of the method more convincing. In particular, there are no comparisons for using no unshared elements (U=0), or the naive approach of fully sharing weights within residual groups.”
>
> Thank you for your suggestion! In our work, having non-shared elements in a filter basis is a trick to avoid the limitations of having low ranks (16 ~64) in small networks (e.g., ResNet56 on CIFAR). Due to low ranks, small networks on CIFAR show low performance when non-shared elements are not used in the shared convolution. For example, on CIFAR-100, when non-shared elements are not used (in ResNet34-S16U0), its accuracy drops by 0.79% from ResNet34-S16U1 that has 1 non-shared element. In the revision, we will update Fig. 5 in Appendix to include results on CIFAR when non-shared elements are not used (when U=0).
>
> Second, fully sharing weights within residual groups implies that sharing both a filter basis and coefficients. Using unshared coefficients to combine a share filter basis is required to improve expression ability. Related work in [1] also demonstrates that sharing coefficients drops 2% more accuracy. According to the reviewer’s suggestion, we conduct further evaluation to investigate the effect of fully sharing weights in residual groups. By fully sharing weights in ResNet50, we can save further 9% parameters (reduced from 18.26M to 16.02M), but its top-1 (top-5) accuracy drops to 75.38% (92.46%), which is 0.67% lower than the counterpart ResNet50. This result shows that having unshared coefficients is important for high performance. We will include this result in the revision.
>
> 2. “the final effect on CIFAR-100 is about 1% accuracy gain in the best case, which is good but seems a little small compared to the motivation of unstable gradients --- might there be other effects, e.g. on the number of steps needed in the training schedule?” and “What happens if orthogonality regularizer is not used on ImageNet? More comprehensive experiments on this would be more convincing.”
>
> Thank you for your suggestion! Both baselines and our models on CIFAR-100 are trained under the same training settings including training schedules, and, hence, we conjecture that the performance gain is obtained by improving the flow of gradients during training.  We conjecture that the effect of orthogonality regularization on final accuracy is not as dramatic as imagined for several reasons. First, parameter-sharing in CNNs is not as extensive as RNNs. Further, since modern networks such as ResNets already have several mechanisms such as skip connections and batch normalization to improve the flow of gradients, the effect of orthogonality regularization can be diluted. According to the reviewer’s suggestion, we conduct further evaluation to demonstrate the effect of orthogonality regularization on ImageNet. Our result shows that when orthogonality regularization is not applied on ResNet50, its top1 accuracy drops to 75.74%, that is 0.41% lower than the counterpart ResNet50. In contrast, when the orthogonality regularization is applied, the same model achieves 76.35%, which is 0.2% higher than the counterpart ResNet50. This 0.61% improvement is obtained simply by applying orthogonality regularization. This performance improvement might seem small at first, but, we believe, it is a big improvement if we consider that ResNet101 has only 1.22% improvement over ResNet50. We believe that this additional result makes our approach more convincing and, hence, we will include this result in the revision.
>
> 3. “What is the effect of lambda in the loss?”
>
> “lambda” is a hyperparameter that determines how orthogonality regularization affects the loss. During evaluation, we have tested lambdas between 0.1 and 1000 and found that our approach is not very sensitive to the value of lambdas and obtained best results when lambda is between 10 and 100. Hence, we use lambda=10 as default during all experiments. We will mention this explicitly in the revision.
>
> 4. “sec 4.4 Fig 3: I don't understand how to interpret this plot, more explanation may be needed. What does the varying lightness of the light blue bars mean --- are there multiple overlapping translucent bars, and why?” and “Also, what does a value of 0.0075 mean --- relative to what? … If gradients are larger or better conditioned, can the training schedule be reduced?”
>
> Fig 3 shows the traces of ‘maximum’ and ‘mean’ gradients flowing in the filter bases during an epoch. On every 10 iterations of a batch, the ‘maximum’ and ‘mean’ gradients are overlapped on top of the old plots. Therefore, the bars look darker if bars are more overlapped.
> The numbers in y-axis are absolute values of the ‘maximum’ and the ‘mean’ gradients recorded at the filter bases. We cannot directly connect “larger or better conditioned” gradients to the reduced training schedule or better performance. However, our evaluation shows that orthogonality regularization and batch normalization have similar effect on the flow of gradients and, hence, we conjecture that orthogonality regularization has similar desirable effects of batch normalization on the performance and training schedule.
>
> 5. “sec 4.4 Fig 4: What are the similarity matrices for groups 1 and 4? Do they show the same pattern? And if not, what might be a cause for difference?”
>
> The coefficients in residual groups 2 and 3 are chosen because they show better contrast when  orthogonality regularization is applied. In our evaluation, orthogonality regularization has less impact on the similarity of coefficients in residual groups 1 and 4. This might be because residual groups 2 and 3 have more repeating convolution blocks than residual groups 1 and 4. For example, in ResNet101, the filter basis for residual group 3 is shared by 22 successive convolution blocks, while residual groups 1 and 4 share their filter bases only twice. Since the parameters are shared more extensively in the residual groups 2 and 3, they have more benefit from orthogonality regularization, resulting in less similarity of the coefficients.
>
> [1] Yang et al. LegoNet: Efficient Convolutional Neural Networks with Lego Filters. ICML 2019.

---

### Official Review · Reviewer_RBVx · 2021-07-17

**Rating:** 5
**Confidence:** 5

**Summary:**

The authors introduce a new parameter sharing scheme in CNNs by learning orthogonal filter bases, and representing each convolutional filters as a linear combination of the bases.
Notably, the authors show both empirically and theoretically that orthogonal regularization imposed on the learned filter bases improves the stability of gradients and helps achieve better overall performance.

**Main Review:**

While learning shared filter basis for parameter reduction in deep neural networks is not a new idea, the discussion on the orthogonality of the bases and its advantages on gradient flow is interesting and constructive.
The introduced method is well presented and the paper is easy to follow, with source code provided for reproduction.
Wide range of experiments including image classification, object detection, and instance segmentation, are appreciated.

I have following concerns:
- As discussed in L81, the authors claim gradient issues cause the performance drop in [1], any support on this?

- Following the previous question, the main contribution of this paper remains a bit unclear to me. If the orthogonal regularization is indeed a principle way of stabilizing the gradient and improve the performance of parameter sharing in CNNs, can it be apply to other methods for improved performance? Or it can only improve the performance of the proposed method?

- How are all the hyperparameters determined? Further ablation study is expected, e.g. the additional orthogonality regularization inevitably introduces new hyperparameter for tuning.

- Missing references and comparisons [2, 3, 4]. These are all published parameter sharing methods for networks and need to be discussed in the paper. Comparisons against [1] are also expected.

- The term 'Deeply' in the proposed method is a bit confusing to me. It seems that the authors are trying to emphasize 'deep', then how is the proposed method different from the previous parameter sharing methods in terms of 'deep'?

[1] Learning implicitly recurrent CNNs through parameter sharing, ICLR 2019
[2] Legonet: Efficient convolutional neural networks with lego filters, ICML 2019
[3] Structured convolutions for efficient neural network design, NeurIPS 2020
[4] HyperNetworks, ICLR 2017


**Time Spent Reviewing:**

3

---

> ### Author Response · Authors · 2021-08-10
> **Thank you for your comment**
>
> 1. “the authors claim gradient issues cause the performance drop in [1], any support on this?” and
>    “can it be apply to other methods for improved performance?”
>
> Although the approaches in [1]~[4] are different in details, they are similar to our work since they all share a set of filters (templates in [1] and lego filters in [2]) and combine them to build  layer-specific filters. However, they save parameters at the cost of accuracy loss. For example, in ResNet50 on ImageNet, [2] and [3] incurs 0.9% and 0.5% accuracy drop, respectively, while our method achieves 0.21% performance improvement. We believe that this performance gap comes from gradient issues of shared parameters and these previous works could benefit more from the improved flow of gradients.
>
> 2. “Comparisons against [1] are also expected.”
>
> In [1], Wide ResNets are used as base models, and, hence direct comparison to our work is not easy. However, our configuration without orthogonality regularization is equivalent to [1] if minor details are ignored.  In order to demonstrate the effect of orthogonality regularization, we have performed further evaluation on ResNet50 on ImageNet. When orthogonality regularization is not used, top-1 accuracy drops to 75.74% (0.41% lower than the counterpart ResNet50.) In contrast, when the orthogonality regularization is applied to the same model, the accuracy is increased to 76.35%, which is 0.2% higher than the counterpart ResNet50. This 0.61% accuracy improvement is achieved simply by applying orthogonality regularization. This performance improvement might seem small at first, but, we believe that it is a big improvement if we consider that ResNet101 has only 1.22% improvement over ResNet50. We believe that this additional result makes our approach more convincing and we will include this result in the revision.
>
> 3. “How are all the hyperparameters determined?”
>
> “lambda” is a hyperparameter that determines how orthogonality regularization affects the loss. During evaluation, we have tested lambdas between 0.1 and 1000 and found that our approach is not very sensitive to the value of lambdas and obtained best results when lambda is between 10 and 100. Hence, we use lambda=10 as default during all experiments. We will mention this explicitly in the revision.
>
> 4. “The term 'Deeply' in the proposed method is a bit confusing to me.”
>
> In related methods [2]-[4], ResNet50 (and Wide ResNet50 in [1]) is used as the deepest model for evaluation. In ResNet50, a residual group has at most 6 iterative convolution blocks, and, hence parameters are shared only a few times. In contrast, we demonstrate the effectiveness of our approach using ResNet101. In our approach, a filter basis is shared by up to 22 successive convolution blocks of ResNet101 while outperforming counterpart ResNet101 by 0.32% in top-1 accuracy. As far as we know, our approach is the only work, at least among [1]~[4],  demonstrating effective sharing of parameter in ‘deep’ networks such as ResNet101 on ImageNet and COCO.

---

### Official Review · Reviewer_fjeu · 2021-07-26

**Rating:** 6
**Confidence:** 3

**Summary:**

In this paper, the authors try to improve the parameter efficiency of neural networks by using recursively shared filter basis. To address the gradient vanishing/exploding problem in naive recursively parameter sharing, orthogonality regularization is proposed and shows effectiveness in ablation studies.

**Limitations And Societal Impact:**

- The position of shared filter basis used in different networks is kind of confusing. It would be great if the authors can explain the reason to not share every convolutions in the blocks (as there're still some non-shared elements in shared conv), maybe also with some ablation studies.
- The parameter reduction in compact models is not very impressive.

**Main Review:**

- The motivation of this paper is very clear. Parameter efficiency is more important in resource-constrained devices like Jetson TX2.
- The proposed parameter sharing approach shows reasonable performance on different benchmarks including ImageNet classification and COCO detection.
- Ablations and analysis are provided to show the effectiveness of proposed orthogonality regularization.

**Time Spent Reviewing:**

4

---

> ### Author Response · Authors · 2021-08-10
> **Thank you for your comment**
>
> 1. “The position of shared filter basis used in different networks is kind of confusing. It would be great if the authors can explain the reason to not share every convolutions in the blocks.”
>
> Thank you for your suggestion! In our work, having non-shared elements in a filter basis is a trick to avoid the limitations of having low ranks (16 ~64) in small networks (e.g., ResNet56 on CIFAR). Due to low ranks, small networks on CIFAR show low performance when non-shared elements are not used in the shared convolution. For example, on CIFAR-100, when non-shared elements are not used (in ResNet34-S16U0), its accuracy drops by 0.79% from ResNet34-S16U1 that has 1 non-shared element. In the revision, we will update Fig. 5 in Appendix to include results on CIFAR when non-shared elements are not used (when U=0).
>
> In contrast, for large networks (ResNet50/101 and MobileNetV2) on large datasets (ImageNet and COCO), we do not use non-shared elements in the filter bases since those filter bases have high ranks between 64 and 512. For these models, we choose one or two convolution operators with the largest parameters for sharing filter bases. In these models, we leave at least one convolution operator as non-shared coefficients to improve expression ability. Related work in [1] also demonstrates that sharing coefficients drops 2% more accuracy. We have conducted further evaluation to demonstrate the effect of sharing parameters for all convolution operators in residual groups. Our results show that we can save further 9% parameters by fully sharing weights in ResNet50, but its top-1 (top-5) accuracy drops to 75.38% (92.46%), which is 0.67% lower than counterpart ResNet50’s. This result shows that having non-shared coefficients is important for high performance. We will include this result in the revision.
>
>
> 2. ”The parameter reduction in compact models is not very impressive.”
>
> Since MobileNet is a compact model, reducing parameters by sharing parameters has fundamental limitations. For example, similar parameter sharing approaches in [1] and [2] save about 17-25% parameters in MobileNets at the cost of 0.9-1.6% accuracy loss. In contrast, our approach reduces 15% parameters at the cost of only 0.01% accuracy loss, and even outperforms the counterpart MobileNetV2 by 0.39% when 8% of parameters is reduced.
>
> [1] Yang et al. LegoNet: Efficient Convolutional Neural Networks with Lego Filters. ICML 2019.
>
> [2] Bhalgat et al. Structured Convolutions for Efficient Neural Network Design. NeurIPS 2020.

---

### Decision · Program_Chairs · 2021-09-27

**Decision:**

Accept (Poster)

**Comment:**

Authors propose to use recursively shared filter basis for compact CNNs. I agree with most reviewers that the idea of shared filter basis in CNN is not new and similar systems have been discussed before. However, ideas such as imposing orthogonality constraint on the shared basis seem interesting, and the effectiveness is well explained. Extensive experiments have been conducted to show improvement.

3 out of 4 reviewers are inclined to accept the paper.